# Aggregation, Cytotoxicity and DNA Binding in a Series of Calix[4]arene Amphiphile Containing Aminotriazole Groups

**DOI:** 10.3390/ph16050699

**Published:** 2023-05-05

**Authors:** Diana Mironova, Egor Makarov, Islamiya Bilyukova, Kevser Akyol, Elsa Sultanova, Vladimir Evtugyn, Damir Davletshin, Elvina Gilyazova, Emil Bulatov, Vladimir Burilov, Svetlana Solovieva, Igor Antipin

**Affiliations:** 1Alexander Butlerov Institute of Chemistry, Kazan Federal University, 18 Kremlevskaya Str., 420008 Kazan, Russia; 2Interdisciplinary Center for Analytical Microscopy, Kazan Federal University, 18 Kremlevskaya Str., 420008 Kazan, Russia; 3Institute of Fundamental Medicine and Biology, Kazan Federal University, 18 Kremlevskaya Str., 420008 Kazan, Russia; 4Arbuzov Institute of Organic and Physical Chemistry, FRC Kazan Scientific Center of RAS, 8 Arbuzov Str., 420088 Kazan, Russia

**Keywords:** calixarene, click chemistry, polyamines, DNA binding, amphiphiles, cytotoxicity

## Abstract

The present work focuses on the study of the aggregation and complexing properties of calixarenes as potential DNA condensation agents for gene delivery. In the current study, 1,4-triazole derivatives of calix[4]arenes **7** and **8** containing monoammonium fragments were synthesized. The synthesized compound’s structure was characterized by using various spectroscopic techniques (FTIR, HRESI MS, ¹H NMR and ¹³C NMR). The interactions between a series of calix[4]arene-containing aminotriazole groups (triazole-containing macrocycles with diethylenetriammonium fragments (**3** and **4**) and triazole-containing macrocycles with monoammonium fragments (**7** and **8**)) and calf thymus DNA were carried out via UV absorption, fluorescence spectroscopy, dynamic light scattering and zeta potential measurements. The role of the binding forces of calixarene–DNA complexes was analyzed. Photophysical and morphological studies revealed the interaction of the calixarenes **3**, **4** and **8** with ct-DNA, which transformed the fibrous structure of ct-DNA to completely condensed compact structures that are 50 nm in diameter. The cytotoxic properties of calixarenes **3**, **4**, **7** and **8** against cancerous cells (MCF7, PC-3) as well as a healthy cell line (HSF) were investigated. Compound **4** was found to have the highest toxic effect on MCF7 breast adenocarcinoma (IC_50_ 3.3 μM).

## 1. Introduction

One of the rapidly developing areas of research in supramolecular chemistry and nanotechnology is the design of systems suitable for numerous applications. Among the wide variety of macrocycles, calixarenes stand out for their ease of functionalization, which allows the introduction of different groups along the upper and/or lower rims [1]. This has led to calixarenes being used in biomedical fields as biocatalysts [2] and anti-tumor agents [3,4], and they have shown excellent antibacterial [5,6,7], antifungal [8,9] and anticancer properties [10,11,12,13]. Calixarenes can exhibit amphiphilic properties when phenolic rings are functionalized with ionic groups and alkyl substituents. This allows calixarenes to establish multiple and cooperative noncovalent interactions—for instance, to transform the deoxyribonucleic acid into a compact form. Thus, the efficiency of several calixarenes as condensing agents has been reported [14,15]. Additionally, calixarenes are excellent host macrocyclic compounds. Their flexible structural properties can help host different guest species, either neutral organics or ionic compounds [16]. Moreover, their capacity to form aggregates allows for the preparation of nanocarriers for different drugs [17,18,19]. Thus, the design of systems based on calixarenes compatible with the biological area and for applications in drug delivery, binding and the detection of biomolecules is an important task. A special place in calixarenes for bioapplications is occupied by aminocalixarenes. They are known to effectively interact with the negatively charged DNA phosphate backbone [20]. It was also reported that the formulation of amphiphilic aminocalix[4]arene-based solid lipid nanoparticles with DNA and chitosan can transfect mammalian cells [21]. According to the data [22], amino-substituted amphiphilic calixarenes have the potential to achieve a targeted delivery of paclitaxel to tumor tissues. Taking into account the relevance of the aggregation properties of calixarenes upon interaction with various biologically active compounds and drugs, in this paper, we report the synthesis of novel water-soluble calix[4]arene amphiphile-containing aminotriazole groups in a preliminary investigation of their potential application in biological fields.

## 2. Results and Discussion

### 2.1. Synthesis

Previously, our research group developed a strategy for the synthesis of tetra-azide derivatives of calix[4]arenes with tetradecyl alkyl chain substitutes **1** and free hydroxyl groups on the lower rim **2** (Figure 1), and triazole-containing macrocycles with diethylenetriammonium fragments **3** and **4** were obtained [23,24].

In this work, to study the effect of the number of amino groups on the binding abilities toward DNA, we synthesized 1,4-triazole derivatives of calix[4]arenes containing four ammonium fragments. For this, macrocycles **1** and **2** were introduced into the reaction of azide-alkyne cycloaddition (CuAAC) with N-propargylphatilimide in toluene (Figure 2) at 40 °C for 24 h using a copper iodide(I)-triethylamine catalytic system. The reaction with the unsubstituted at the lower rim macrocycle **2** was carried out with the addition of DMF to increase the solubility of the initial substrate. Target products **5** and **6** were isolated in 70% and 89% yields, respectively.

In the ^1^H NMR-spectra of compound **5**, the proton signals of the triazole and aromatic fragments appeared as two singlets at 7.75 and 7.01 ppm, respectively (Appendix A). Two doublets of doublets at 7.80 and 7.68 ppm with the coupling constants *J* = 3.1 and 5.4 Hz, respectively, corresponded to the proton signals of the phthalimide substituent, and a singlet at 5.01 ppm belonged to the methylene group, bridged to the triazole fragment. The protons of the tetradecyl substituent appeared at 3.94 and 0.87 (triplets), 1.91 (multiplet) and 1.66 and 1.38 (broadened singlets). The successful introduction of the phthalimide fragment into the structure of macrocycle is evidenced by the appearance of an absorption band of the C=O group at 1717 cm^−1^ in the FTIR-spectra (Appendix A) and in the ^13^C NMR-spectra of the signal of its carbonyl carbon at 167.80 ppm (Appendix A). The presence in HRESI MS-spectra [M+H]^+^, [M+2H]^2+^ ions with *m/z* 2115.2503 and 1058.1295, respectively (theoretical 2115.2508 and 1058.1290, respectively) unambiguously prove the composition of **5.** The appearance of the proton signals of the triazole and phthalimide fragments at 8.39 and 7.87 ppm as a singlet and a multiplet, respectively, in the ^1^H NMR spectra (Appendix A) indicates the conversion of the target product **6**. The proton signals of the calixarene platform appeared as two singlets at 7.61 (aromatic) and 3.55 ppm (bridged methylene groups); for the CH_2_-group between triazole and phthalimide, it appeared at 4.84 ppm. In the FTIR spectra (Appendix A), similar to compound **5**, the absorption band of the C=O group at 1712 cm^−1^ was discovered. HRESI MS-spectra (Appendix A) in a positive regime showed the ion [M+H]^+^ with *m/z* 1329.3704 (theoretical 1329.3710), which confirms the structure of **6**.

Phthalimide protection was removed by reacting macrocycles **5** and **6** with hydrazine hydrate in methanol. After treatment with hydrochloric acid solution and subsequent precipitation, calix[4]arenes **7** and **8** were isolated in 72% and 90% yields, respectively.

In the ^1^H NMR-spectra (Appendix A) of compound **7**, the composition of the target product is evidenced by the disappearance of signals from the protons of the phthalimide fragment. The strong broadening of all proton signals was due to the aggregation of the macrocycle in deuterosolvents. Singlets at 7.71 and 7.07 ppm corresponded to the proton signals of triazole and aromatic rings, respectively. In addition, the signals of N-CH_2_ and O-CH_2_ protons overlapped in the spectra, which led to a broadened singlet at 3.97 ppm with an integral intensity equal to ~16. The remaining signals were correlated according to the structure of calixarene. Taking into account four amino groups in the molecule of macrocycle **7**, multiply charged ions were observed in the HRESI MS-spectra (Appendix A) in a positive regime: [M+H]^+^, [M+2H]^2+^ and [M+3H]^3+^ with *m/z* 1595.2221, 798.1186 and 532.4148, respectively (theoretical 1595.2289, 798.1181 and 532.4151, respectively).

The ^1^H NMR-spectra of compound **8** (Appendix A) were registered in deuterium oxide due to its great solubility in water. Peaks were found in the spectra at 8.27, 7.49 and 4.21 ppm, which were related to the proton signals of the triazole ring, aromatic fragment and aminomethylene group, respectively. The signals of the bridged methylene groups overlapped with the signal of the solvent and appeared as a singlet, which is consistent with the general theory for macrocycles unsubstituted at the lower rim: the polar solvent participates with OH-groups in the formation of hydrogen bonds, which leads to an increase in the conformational mobility of calixarene in solution [25]. For the same reason, the carbon signals in the ^13^C NMR spectra (Appendix A) were strongly broadened. Even lowering the temperature to 5 °C did not narrow the signals. In the FTIR-spectra, a broad absorption band at 2925 cm^−1^ was characteristic, reflecting the stretching vibrations of OH-, CH_ar_ and NH_3_^+^-groups (Appendix A). Despite the presence of four charged amino groups in the structure of macrocycle **8**, only ions [M+H]^+^ and [M+2H]^2+^ with *m/z* 809.3482 and 405.177, respectively (theoretical 809.3491 and 405.1782, respectively) were observed in the mass-spectra (Appendix A).

### 2.2. Study of the Self-Assembly of Calixarenes

#### 2.2.1. Determination of the Critical Aggregate Concentration (CAC)

The CAC of calixarenes was determined by using pyrene as a probe [26] (Appendix A). The values of CAC for the calixarenes studied are listed in Table 1. The CAC values for the **4** and **8** calixarenes cannot be determined, as the structures of these macrocycles lack lipophilic fragments that contribute to spontaneous association into aggregates. For **3**, the CAC value was slightly higher than for **7**. The prevention of the early aggregation of **3** compared to **7** was due to the larger volume and total charge of hydrophilic groups in **3**.

#### 2.2.2. Determination of the Size and Zeta Potential of Calixarenes

In Table 1, the average diameters of the nanostructures measured by using the dynamic light scattering method (DLS) are presented. Particle sizes at concentrations above CAC (10 μM) were determined. In addition, in Table 1, the polydispersity indices (PDI) for studied systems are assembled. Despite the bulk hydrophilic groups on the upper rim, **3** at concentrations above CAC predominantly formed aggregates with an average hydrodynamic diameter of 20 nm.

### 2.3. Interaction of Calixarenes with DNA

#### 2.3.1. Absorption Spectroscopy Measurements

The UV-vis absorption spectrum of ct-DNA with absorption maximum at 260 nm reveals a broad band (Figure 1). As a result of conformational changes in the DNA structure upon interaction with calixarenes’ red/blue shift, hyperchromic/hypochromic effects in UV-vis spectra could be observed. The addition of **7**, **4** and **3** to ct-DNA solution led to an increase in the hyperchromic effect. For **8**, a hypochromic effect and a slight bathochromic shift of the calixarene absorption maximum were observed. For **3**, the maximum hyperchromic effect was fixed. Therefore, ct-DNA double strands were folded simultaneously due to assembly interactions, such as hydrophobic effects and hydrogen bonds between the corresponding bases. Thus, the hyperchromic effect was due to the presence of a charged cationic fragment of the calixarene, which interacted with the phosphate group of the DNA backbone through the Coulomb interaction.

#### 2.3.2. DNA Melting Temperature (T_m_) Study

The temperature increases lead to the denaturation of dsDNA into single-stranded DNA (ssDNA) due to the weakening of base and hydrogen bond interactions. Changes in the structure of the DNA molecules led to changes in their conformation and physical properties (absorption and melting point). The DNA melting temperature (T_m_) is defined as the temperature at which half of the double-helical DNA structure is unfolded to two individual strands [27]. On one hand, strong intercalation can stabilize the double helix of DNA, resulting in a 5–10 °C increase in T_m_. On the other hand, for groove or Coulomb binding, the change in T_m_ is negligible [28]. The T_m_ values were calculated from the melting curve (Figure 2). For free ct-DNA, T_m_ was 73.1 °C, whereas in the presence of calixarenes **3**, **4**, **7** and **8**, it was found to be 84.1 °C, 75.2 °C, 74.1 °C and 75.0 °C, respectively. Strong changes in T_m_ values for ct-DNA in the presence of poly amino derivatives of calixarene (**3** and **4**) indicate the strong intercalation between calixarenes and ct-DNA.

#### 2.3.3. Fluorescence Spectroscopic Studies—Competitive Displacement Assay

The competitive binding of EtBr (ethidium bromide) and calixarenes to ct-DNA was used to quantify interactions between DNA and the calixarenes by using fluorescence emission spectroscopy. The EtBr fluorescence intensity increased significantly when intercalated into base pairs of ct-DNA. However, the addition of calixarenes displaced the EtBr from DNA, which resulted in fluorescence quenching. Calixarenes displaced EtBr from the ct-DNA in a concentration-dependent manner but with various curve profiles. The concentration increased for calixarenes with alkyl substituents on the lower rim (**3** and **7**) in DNA solution, resulting in a smooth quenching fluorescence emission from EtBr (Figure 3c,d). Additionally, the **3** and **8** in DNA solutions in the range of 10–15 μM led to strong changes in the quenching of the EtBr emission (Figure 3a,b). This phenomenon was associated with a change in the mechanism of interaction between calixarene and DNA. It may have been the result of the formation of DNA/calixarene complexes other than in a 1:1 ratio. The formation of complexes of different compositions led to a redistribution of electron density, which in turn strongly changed the quantum yield of the fluorescence emissions.

For the quantification of intermolecular interactions between calixarenes with EtBr-DNA systems, the well-known Stern–Volmer plots were used according to Equation (1):(1)$$\frac{F_{0}}{F}=1+K_{SV}\left[Q\right]=1+k_{q}+\tau_{0}[Q],$$
where *F*_0_ and *F* represent the emission intensities of EtBr-DNA without and with calixarenes, respectively, and *K_SV_* represents the quenching constant. $k_{q}$ and $\tau_{0}$ are the bimolecular quenching rate constant and fluorescence lifetime without quenching ($\tau_{0}$ = $10^{-8}$ s), respectively. The evaluated quenching constant ($K_{SV}$ values) also designates that calixarenes **3**, **4** and **8** bind to ct-DNA (Table 2). According to Equation (1), *K_q_* is higher than the limiting diffusion rate constant ($2\times 10^{10}M^{-1}s^{-1}$) for a biomacromolecule, indicating the existence of a static quenching mechanism [29].

The binding constant (*K_b_*) and the stoichiometry of binding (n) per DNA molecule were computed from the intercept and slope of the plot of $log(F_{0}-F)/F$ versus log (calixarene), as shown in (Figure 4b) and Table 2, using Equation (2):(2)$$log\frac{F_{0}-F}{F}=logK_{b}+nlog\left[C\right],$$
where $K_{b}$ is the binding constant and n is the number of binding sites. For all studies of systems and calixarenes in DNA solution, except **7**, the binding constants and number of bindings cites were close to each other. The results obtained for **3**, **4** and **8** in DNA solution indicates the same mechanisms of interaction realized in studied systems in a low concentration range. In the case of **7**, the estimation of the binding constant with DNA is not possible due to the low-quality dependence between the emission fluorescence of DNA and the concentration of calixarene. The reason for this can be related to the strong self-association of **7**, which prevents its interaction with DNA.

The binding force between protein and ligand complexes is noncovalent bonds, which can be divided into hydrogen bonds, hydrophobic forces, Columbic interactions and van der Waals interactions. For estimation of the thermodynamic parameters, a linear van’t Hoff plot between binding constants and inverse temperature was applied (see Appendix A).
(3)$$lnK=\frac{H}{RT}+\frac{S}{R}$$
(4)ΔG=ΔH−TΔS,

From the van’t Hoff plot, the thermodynamics parameters of the binding ligand–protein complexes were calculated, and they are assembled in Table 3. The change in the negative value of free energy suggests that the interaction between calixarenes and DNA was spontaneous and feasible. The negative values of enthalpy change explored that the van der Waals interaction along with the hydrogen bonds were the major binding forces involved in the protein–ligand complexes [30].

#### 2.3.4. Particle Size and Zeta Potential Measurements

The dynamic light scattering method (DLS) is a common technique for studying changes of particle size and zeta potentials of DNA complexes with calixarenes at different concentrations. In Figure 5, the obtained results are presented. The addition of calixarenes **3**, **4** and **8** to DNA at a calixarene concentration of 5 μM led to the compactization of DNA from 562 nm to 150–100 nm and low values of the polydispersity index (Appendix A). Previously [24], it was shown that the addition of calixarenes **3** and **4** to DNA does not cause changes in the native DNA structure, according to the circular dichroism (CD) method. The addition of calixarene **7** to DNA did not significantly decrease the average hydrodynamic diameter of the calixarene–DNA double system, even at an equimolar ratio of 1:1 (50 μM) (Figure 5a).

The addition of calixarenes **3**, **4** and **8** caused the gradual neutralization of negative DNA charges (Figure 5b). Thus, the calixarenes led first to electrostatic binding, and in the case of **3**, to DNA immersion into aggregates, which caused the increase of the zeta potential of calixarene–DNA systems to positive values. The addition of calixarene **7** to DNA practically had no effect on the zeta potential of DNA.

#### 2.3.5. Transmission Electron Microscopy (TEM)

The surface morphology of the calixarene–DNA systems was observed by using TEM. In Figure 6, the structures of free DNA and DNA in the presence of calixarenes are presented. Images of free DNA show typical fibril-like structures [31]. The highly condensed structures generated after the addition of calixarenes in DNA solution are shown in Figure 6b–d.

In Figure 6, the formation of gorgon-like aggregates between **4** and **8** with DNA was found. This indicates that **4** and **8** do not form intramolecular DNA condensates. However, simultaneously, the interaction of **4** and **8** with several DNA strands could be what generated the observed gorgon-like aggregates (Figure 6b,c). In this case, electrostatic interactions between the positively charged upper rim groups of **4** and **8** and negatively charged DNA bases are dominant. For lipophilic **3**, the compactization of DNA into vesicle-like aggregates was observed. For calixarene **7**, no usable samples could be obtained.

### 2.4. Cytotoxicity Analysis of Calixarenes

During the dissolution of the compounds, we found that compound **7** was poorly soluble in water up to a concentration of 20 mM, unlike other studied compounds. Therefore, we decided to decrease the concentration of compound **7** to 5 mM (although the formation of clots was observed, which dissolved upon heating to 100 °C). In addition, during the serial dilution of compounds in the culture medium, the formation of insoluble precipitates was observed. They were especially clearly visible at concentrations of 320 µM and 640 µM. These clots gave a false positive result during the MTT-test. Therefore, it was not possible to reliably determine the degree of influence of the studied compounds at concentrations of 320 µM and 640 µM. Thus, it was decided not to take into account these concentrations, and therefore, the concentration range shown on the graph is up to 160 µM (Figure 7).

We found that calixarene **4** exhibited greater cytotoxic activity against MCF7 breast adenocarcinoma and PC-3 prostate carcinoma (Table 4). The minimal cytotoxicity of calixarene **4** against normal HSF cell lines was observed. The values in the semimaximal inhibition range for calixarene **4** from 3 µM to 109 µM. Calixarene **3** exhibited cytotoxic activity against MCF7 breast adenocarcinoma at the same values as for calixarene **4** (Table 4). However, for calixarene **3**, cytotoxicity against normal HSF cell lines was also observed. For compound **7**, cytotoxic activity was also observed against the PC-3 cell line (IC_50_—108 µM) (Table 4). The IC50 values for compound **8** could not be determined within the studied concentration range. The results obtained for calixarene **4** correlate with the IC_50_ values obtained for the amine-containing calix[4]arene [32]. However, the presence of triazole groups on the upper rim and OH groups on the lower rim allowed for increased selectivity towards MCF7 compared to healthy cells. Nevertheless, more studies are needed to confirm this lack of selectivity and to compare its effects with anticancer agents in clinical use.

## 3. Materials and Methods

### 3.1. Characterization Methods

^1^H and ^13^C NMR spectra as well as 2D ^1^H-^1^H NOESY were recorded on Bruker Avance 400 Nanobay (Bruker Corporation, Billerica, MA, USA) with signals from residual protons of CDCl_3_, D_2_O or DMSO-d_6_ as the internal standard.

The melting points were measured using the Optimelt MPA100 melting point apparatus (Stanford Research Systems, Sunnyvale, CA, USA).

IR spectra in KBr pellets were recorded on a Bruker Vector-22 spectrometer (Bruker Corporation, Billerica, MA, USA).

High-resolution mass spectra with electrospray ionization (HRESIMS) were obtained on an Agilent iFunnel 6550 Q-TOF LC/MS (Agilent Technologies, Santa Clara, CA, USA) in positive or negative mode with the following parameters: nitrogen carrier gas; temperature, 300 °C; carrier flow rate, 12 L × min^−1^; nebulizer pressure, 275 kPa; funnel voltage, 3500 V; capillary voltage, 500 V; total ion current recording mode; 100–3000 *m/z* mass range; scanning speed, 7 spectra × s^−1^.

### 3.2. Materials

Materials of the highest purity available were purchased from Fluka and Sigma-Aldrich and used as received. Solvents were purified according to standard methods. 3-Bis[2-(tertbutoxycarbonylamino)ethyl]propargylamine, 25,26,27,28-tetrahydroxycalix[4]arene, 5,11,17,23-tetraazo-25,26,27,28-tetrahydroxycalix[4]arene, 5,11,17,23-tetraamino-25,26,27,28-tetrahydroxycalix[4]arene [33,34,35] and 5,11,17,23-tetraazido-25,26,27,28-tetrahydroxycalix[4]arene, 5,11,17,23-tetraazido-25,26,27,28-tetratetradecyloxycalix[4]arene [23,24] were prepared following procedures in the literature.

#### Synthesis

The 5,11,17,23-tetra(4-phtalimidomethyl-1,2,3-triazol-1-yl)-25,26,27,28-tetratetradecyloxycalix[4]arene (**5**), synthesis of 5,11,17,23-tetra(4-phtalimidomethyl-1,2,3-triazol-1-yl)-25,26,27,28-tetrahydroxycalix[4]arene (**6**), 5,11,17,23-tetra(4-aminomethyl-1,2,3-triazol-1-yl)-25,26,27,28-tetratetradecyloxycalix[4]arene tetrahydrochloride (**7**) and 5,11,17,23-tetra(4-aminomethyl-1,2,3-triazol-1-yl)-25,26,27,28-tetrahydroxycalix[4]arene tetrahydrochloride (**8**) were synthesized according to previously developed methods with minor modifications [24]. Detailed descriptions of the synthesis procedures are present in the Appendix A.

### 3.3. Sample Preparation

All solutions were prepared with Adrona Crystal E30 MilliQ water, resistivity 0.055 microSiemens (SIA «Adrona», Latvia, Riga). The highly concentrated solutions of macrocycles (1 mM) and EtBr (1 mM) were prepared in water. The pH levels of the solutions were maintained at 7.4 by using 10 mM TRIS-HCl (tris(hydroxymethyl)aminomethane) buffer. Amounts of ct-DNA were dissolved in 10 mM Tris-HCl buffer (pH 7.4) and kept at 4 °C for 24 h. Calf thymus DNA concentration was evaluated by using a previously developed approach [36].

### 3.4. UV Absorption Spectroscopic Study

The UV-Vis spectra were recorded by using spectrophotometer Shimadzu UV-2600 (Shimadzu, Kyoto, Japan). The spectra were recorded between 240 and 320 nm using a quartz cuvette of path length 1 cm.

### 3.5. Steady-State Fluorescence Study

The emission spectra were recorded by using spectrofluorometer Jobin Yvon Horiba Fluorolog-3 (HORIBA Jobin Yvon SAS, France). Fluorescence emissions were registered in the range of 535–750 nm with excitation at 525 nm and a 3 nm slit. The CAC values were measured according to the method presented in [24].

### 3.6. Dynamic and Electrophoretic Light Scattering Study

DLS and ELS measurements were carried out on a Zetasizer Nano ZS instrument (Malvern Panalytical, UK) with a 4 mW 633 nm He–Ne laser light source and a light scattering angle of 173 degrees. The solutions were filtered through an 800 nm filter before measurements to remove dust.

### 3.7. Transition Electron Microscopy (TEM) Study

TEM was performed on a Hitachi HT7700 ExaLens (Hitachi High-Tech Corporation, Tokyo, Japan). Sample preparation was performed using a previously developed method [24].

### 3.8. Cell Culture Cultivating

Tumor cell lines MCF7 (breast adenocarcinoma) and PC-3 (prostate carcinoma) were cultured in DMEM media with the addition of 5% inactivated fetal bovine serum, 1 mM L-glutamine and a mixture of penicillin (5000 U/mL) and streptomycin (5000 µg/mL). The HSF cell line (human skin fibroblasts) was cultured in DMEM media with the addition of 10% inactivated fetal bovine serum, 1 mM L–glutamine and a mixture of penicillin (5000 U/mL) and streptomycin (5000 µg/mL). The cells were cultured at 37 °C in a humid atmosphere containing 5% CO_2_.

The cells were placed in 96-well culture plates with a flat bottom at a concentration of 4 × 10^3^ cells per well (for PC-3 and MCF7) and 5 × 10^3^ cells per well (for HSF) in a full culture medium and incubated at 37 °C with 5% CO_2_ for 24 h.

### 3.9. Addition of the Studied Compounds to Cells

Stock solutions of the compounds were prepared before the experiment by dissolving the compounds in deionized water to final concentrations of 20 mM (**3**, **4** and **8**) and 5 mM (**7**). The compounds were added to the wells with cells to obtain final concentrations of 5 µM, 10 µM, 20 µM, 40 µM, 80 µM, 160 µM, 320 µM and 640 µM. Equal volumes of medium were added to untreated cells (control).

### 3.10. Cytotoxicity Analysis of the Compounds

The cytotoxicity of the compounds in tumor cell lines was analyzed using a colorimetric MTT-test, and the results were detected using microplate reader Infinite M200 (Tecan, Männedorf, Switzerland). The cytotoxicity of the studied compounds was analyzed in 3 technical and 2 biological replicates.

The cytotoxicity of the studied compounds was determined using the MTT assay, a colorimetric assay for assessing the metabolic activity of cells. Briefly, yellow tetrazole was reduced to insoluble purple formazan in living cells. A solvent (usually dimethyl sulfoxide) was added to convert the insoluble purple formazan to a colored solution. The absorbance of this colored solution can be expressed quantitatively by measuring it at a specific wavelength (usually between 500 and 600 nm) using spectrophotometry.

MTT reagent was added at the end of the experiment to analyze the number of live cells. The incubation of cells with MTT reagent took 4 h, whereas the division cycle of a tumor cell was ~12 h. Therefore, the effect of MTT on the proliferative activity of cells was insignificant. A more detailed description of the experimental procedure is present in the Appendix A.

## 4. Conclusions

In this work, triazole-containing calix[4]arene with monoammonium fragments was synthesized. The ct-DNA binding, subsequent DNA condensation and cytotoxicity studies of series of calix[4]arene amphiphile-containing aminotriazole groups using optical, morphological and MTT assay techniques was demonstrated. The results of the DLS and TEM methods indicate the process of DNA condensation. The role of positively charged upper rim groups of calixarenes in controlling the process of DNA condensation was confirmed by using zeta-potential measurements. It is shown that electrostatic interaction and intercalation induces efficient DNA condensation via calixarene molecules. The dominant contribution of hydrogen bonds in the complex formation between calixarene and DNA is established. The in vitro biological applicability of calixarene-containing aminotriazole groups tested by performing an MTT assay on cancer and normal cells revealed greater toxicity to cancer cells than to healthy cells. The present study contributes to a better understanding of the factors affecting DNA macrocycle binding and may help to develop more effective and safer vectors for gene therapy.

## Data Availability

Data is contained within the article or the Appendix A.

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
