# Peer review of "Aggregation, Cytotoxicity and DNA Binding in a Series of Calix[4]arene Amphiphile Containing Aminotriazole Groups"

_pharmaceuticals, 2023, doi:10.3390/ph16050699_

Round 1

Reviewer 1 Report

Mironova et al. presented their work focused on studying the aggregation and complexing properties of calixarenes as potential DNA condensation agents for gene delivery. They synthesized the compound’s structure of 1,4-triazole derivatives of calix[4]arenes 7 and 8, containing monoammonium fragments  and characterized using various spectroscopic techniques (FTIR, HRESI MS, ¹H NMR, and ¹³C NMR). The authors also determined cytotoxic properties of calixarenes 3, 4, 7, 8 against cancerous cells (MCF7, PC-3)and compound 4 was found to have the highest toxic effect on MCF7 breast adenocarcinoma cells. The study is very novel and will draw lot of attention from the viewers. I have some minor concerns:

1. Some of the words are misspelled. For e.g, supplementary. Please check thoroughly.

2. How many times the cytotoxicity study was performed? Please add. 

3. When the authors state cytotoxicity, please mention whether they used any dye to quantify it. For example, there is cytox (red/green) dye. Is the cell showing less proliferation with those compounds.

4. Can the authors perform an invasion assay using compound 4 in MCF7 breast adenocarcinoma cells?

Quality of science is great, but the language could be improved. Please check spellings and typos.

Author Response

We thank Editor and Reviewers for comments and suggestions for improving the manuscript.

Reviewer #1. 1. Some of the words are misspelled. For e.g, supplementary. Please check thoroughly.

Response: The manuscript text was corrected.

Reviewer #1. 2. How many times the cytotoxicity study was performed? Please add. 

Response: The cytotoxicity of the studied compounds was analyzed in 3 technical and 2 biological replicates.

Reviewer #1. 3. When the authors state cytotoxicity, please mention whether they used any dye to quantify it. For example, there is cytox (red/green) dye. Is the cell showing less proliferation with those compounds.

Response: The cytotoxicity of the studies compounds was determined using the MTT assay, a colorimetric assay for assessing the metabolic activity of cells. MTT - yellow tetrazole, is reduced to insoluble purple formazan in living cells. A solvent (usually dimethyl sulfoxide) is added to convert the insoluble purple formazan to a colored solution. The absorbance of this colored solution can be expressed quantitatively by measuring at a specific wavelength (usually between 500 and 600 nm) using spectrophotometry.

MTT reagent is added at the end of the experiment to analyze the number of live cells. Incubation of cells with MTT reagent takes 4 hours, while the division cycle of a tumor cell is ~ 12 hours. Therefore, the effect of MTT on the proliferative activity of cells is insignificant.

Reviewer #1. 4. Can the authors perform an invasion assay using compound 4 in MCF7 breast adenocarcinoma cells?

Response: In our experiment the cytotoxic effect of the compounds suggests that they enter the cell. The result of the MTT test is cell death, and this fact suggests that the compounds under investigation are capable of penetrating the cells.

Reviewer 2 Report

Comments:

The original paper by Diana Mironova and co-workers synthesized 1,4-triazole derivatives of calix[4]arenes and studied their aggregation and complexing properties of as potential DNA condensation agents for gene delivery. The novelty and importance of the research are recognized. However, some concerns are suggested to take into consideration and addressed in the text.

Questions:

1.     The correlation coefficient (R2) are suggested to more accurate to four decimal places.

2.     What was the temperature for the emission spectra test? The effect of the temperature should be considered in fluorescence spectroscopic studies section.

3.           The binding constant (Kb) is related to the thermodynamic parameters ΔG, ΔH

and ΔS. The thermodynamic analysis is suggested to added.

Based on the content and quality of this paper, I think this work is suitable for Pharmaceuticals. However, the above problems should be addressed before accepting this manuscript for publication.

Author Response

We thank Editor and Reviewers for comments and suggestions for improving the manuscript.

Reviewer #2. 1. The correlation coefficient (R2) are suggested to more accurate to four decimal places.

Response:  In Table 2 the correlation coefficient data were corrected.

Reviewer #2. 2. What was the temperature for the emission spectra test? The effect of the temperature should be considered in fluorescence spectroscopic studies section.

Response: All fluorescence specter were registered at 298.15K. The temperature dependences of fluorescence specter were added in the manuscript text for thermodynamics analysis.

Reviewer #2. 3The binding constant (Kb) is related to the thermodynamic parameters ΔG, ΔH and ΔS. The thermodynamic analysis is suggested to added.

Response: The thermodynamic analysis part was added in the manuscript text.

The binding force between protein and ligand complexes is non-covalent bonds which can be divided into: hydrogen bond, hydrophobic forces, Columbic interactions and van der Waals interactions. For estimation the thermodynamic parameters a linear van ‘t Hoff plot between binding constants and inverse temperature were applied (see Fig.SX).

Form van ‘t Hoff plot thermodynamics parameters of binding ligand-protein complexes were calculated and assembled in Table 3. The change in the negative value of free energy suggested that the interaction between calixarenes and DNA was spontaneous and feasible. The negative values of enthalpy change explored that the van der Waals interaction along with hydrogen bonds were the major binding force involved in the protein–ligand complexes. [30]
